# Flexural Creep Response of Hybrid GFRP–FRC Sandwich Panels

**DOI:** 10.3390/ma15072536

**Published:** 2022-03-30

**Authors:** Tiago Silva, Luís Correia, Mozhdeh Dehshirizadeh, José Sena-Cruz

**Affiliations:** 1Institute for Sustainability and Innovation in Structural Engineering (ISISE)/Institute of Science and Innovation for Bio-Sustainability (IB-S), University of Minho, Azurém, 4800-058 Guimarães, Portugal; id10078@alunos.uminho.pt (T.S.); lcorreia@civil.uminho.pt (L.C.); 2Department of Civil Engineering, Yazd University, Yazd 8915818411, Iran; mozhdeh.dehshirizadeh@gmail.com

**Keywords:** GFRP, FRC, hybrid sandwich panels, flexural behaviour, creep, analytical modelling

## Abstract

This work was developed within the scope of the research project “Easyfloor—Development of composite sandwich panels for building floor rehabilitation”, which aims at developing an innovative hybrid sandwich panel as an alternative construction system to conventional floor solutions, mainly for building rehabilitation. The developed hybrid sandwich panel is composed of a top face layer of steel-fibre-reinforced self-compacting concrete (FRC), a core of polyurethane (PUR) closed-cell foam, a bottom face sheet, and lateral webs of glass-fibre-reinforced polymer (GFRP). Full-scale experimental tests on the developed sandwich panels were carried out to characterize their short- and long-term (creep) flexural behaviour. The present work includes a detailed description of the developed panels and the experimental programme carried out and presents and discusses the relevant results. The experimental results showed an almost linear behaviour up to failure. The creep tests were carried out for a period of 180 days, using a creep load equal to 20% of its ultimate loading capacity. An average creep coefficient of 0.27 was obtained for this period. The composed creep model used to simulate the sandwich panel’s creep deflections by considering the individual viscoelastic contributions was able to predict the observed structural response with good accuracy.

## 1. Introduction

Fibre-reinforced polymer (FRP) sandwich panels have been successfully applied in civil engineering structural applications due to their light weight, high resistance, high stiffness-to-weight ratio, and enhanced durability [1,2,3], namely as structural elements in bridge decks [4] and roof structures [5]. Nevertheless, their brittle failure and lack of specific design codes have so far hindered the widespread use of these FRP panels [6,7,8]. In order to overcome some of these disadvantages, which often lead to instability (local and global buckling), several researchers have proposed hybrid GFRP–concrete structural solutions [9,10,11,12,13,14,15,16,17,18]. Furthermore, the top layer of concrete provides higher ductility, fire endurance, and impact resistance. Most of the previously mentioned investigations reported the short-term static behaviour of hybrid sandwich panels. They also developed useful methods to predict the short-term responses of hybrid structures.

To the best of the authors’ knowledge, there are only three studies in the literature about the creep response of hybrid GFRP–concrete structures [19,20,21]. In the first two studies [19,20], the creep behaviour of hybrid beams made of GFRP pultruded I-section profiles connected to a thin steel-fibre-reinforced self-compacting concrete (SFRSCC) layer by M10 steel anchors and a thick epoxy adhesive layer was investigated, while the in a later study GFRP I-profile was adhesively bonded (with an epoxy adhesive) to a reinforced concrete slab. Mendes et al. [19] tested two 6.0-metre-long bridge prototypes under a uniform bending load during 21 and 51 days. The difference between the two prototypes was limited to the longitudinal distance between the M10 anchors, which was 200 mm in the one tested for 21 days (PROT_200), and 125 mm in the other (PROT_125). The results obtained showed that the long-term deflection of prototypes was not affected by using a larger distance between the M10 anchors. However, at the end of 51 days of loading, the deflection in PROT_125 increased by 60% due to the creep effect. Moreover, the authors performed analytical predictions adopting the creep model proposed by Bank [22] for GFRP profiles and Eurocode 2 [23] and CEB-FIP Model Code 1990 [24] for the SFRCC layer. In this study, the analytical predictions used did not explicitly consider the shear deformability (instantaneous or long-term), which is generally not negligible in GFRP members [22].

Gonilha et al. [20] carried out experimental and analytical studies on the creep behaviour of a 6.0-metre-long footbridge prototype. The footbridge prototype was tested under a uniform bending load for time durations up to 2642 h in three different combinations of load levels and environmental conditions. The experimental results showed that environmental conditions, namely temperature and relative humidity, significantly influence the viscoelastic response of hybrid structures. Moreover, the results showed that GFRP–concrete hybrid structures lead to a considerable decrease in the creep deflection of GFRP structures. The midspan deflection of the footbridge was predicted by proposing a composed creep model based on the Timoshenko beam theory, which considers the shear deformability of the footbridge during the creep tests. The creep model proposed by Bank [22] and/or by EuroComp [25] was used to predict the time-dependent flexural and shear moduli of GFRP profiles, whereas for the SFRSCC deck, the creep model proposed by Eurocode 2 [23] was used to calculate the viscoelasticity modulus. The composed creep model proposed by the authors was able to accurately predict the experimental results. It has been shown that for long-term predictions, Findley’s power law regressions of short-term experimental tests diverge considerably from the analytical predictions. The results of the above-mentioned tests were then used to design a full-scale 11-m-long footbridge [26].

Recently, Alachek et al. [21] investigated the creep behaviour of three simply supported hybrid beams with a GFRP I-profile connected to a reinforced concrete slab by an epoxy adhesive. The beams were tested under a uniform bending load for up to 3500 h in a natural environmental condition. The authors found that the creep and shrinkage of the concrete, in addition to the creep of the GFRP profile, increased the deflection of the hybrid beams; therefore, it was suggested that the effect of environmental conditions be considered when designing hybrid structures connected by an epoxy adhesive. Moreover, a finite element analysis was performed to simulate the long-term behaviour of the hybrid beam. A good agreement was found between the numerical simulation and the experimental results.

In this study, a new hybrid sandwich panel was developed within the scope of the Easyfloor R&D project. Therefore, experimental and analytical investigations were carried out on the short and long-term behaviour of this hybrid GFRP–FRC sandwich panel. The experimental programme included full-scale (i) flexural tests with variable span (VST), (ii) flexural tests up to failure (FFT), and (iii) flexural creep tests (FCT) on hybrid GFRP–FRC sandwich panels, each one with a length of 4.7 m and a rectangular cross-section of 300 mm by 160 mm. The panels have (i) face sheets and lateral webs of pultruded glass-fibre-reinforced polymer (GFRP), (ii) a core of polyurethane (PUR) closed-cell foam, and (iii) a top layer made of steel-fibre-reinforced self-compacting concrete (FRC). Creep tests were performed by applying uniformly distributed gravity loads over the span of the panels for a period of 2203 h and measuring the midspan deflections and strains over time. In the analytical part of this study, the long-term response of hybrid GFRP–FRC panels is predicted by using Findley’s power law and a composed creep model (CCM) based on the creep response of the constituent materials. The composed creep model is based on the Timoshenko beam theory considering the creep models proposed by Bank [22] and/or by EuroComp [25] to predict the time-dependent flexural and shear moduli of GFRP profiles, as well as Eurocode 2 [23] to calculate the viscoelasticity modulus of the FRC layer.

## 2. Materials, Hybrid Sandwich Panel, Experimental Programme and Test Methods

### 2.1. Materials and Characterization

The hybrid GFRP–FRC sandwich panel studied in this work is composed of (i) a top layer in steel-fibre-reinforced self-compacting concrete (FRC), and (ii) a glass-fibre-reinforced polymer (GFRP) pultruded panel with (iii) a core of polyurethane (PUR) closed-cell foam (see Figure 1). To promote the bond between the FRC layer and the GFRP, (iv) an epoxy adhesive was used upon the casting of the fresh wet FRC. The materials composing the hybrid sandwich panel were experimentally characterized and the results are described in the following paragraphs.

The main characteristics of the adopted FRC were derived from (i) the requirements obtained in the preliminary studies on the optimization of the sandwich panel (see more details in Section 2.2), mainly the required mechanical properties, and from (ii) the needs at fresh state, namely self-compacting features.

Table 1 presents the composition of the FRC. The fibre-reinforced concrete was composed of Portland cement I 42.5 R, fly ash, coarse and crushed aggregates, fine sand, superplasticizer (SK-617), steel fibres (length: 33 mm; slenderness ratio: 60), and polypropylene fibres (length: 12 mm).

Two FRC batches (B1 and B2) were used to cast all the studied panels, and their characterization was carried out simultaneously with the flexural tests up to failure of the panels. The mechanical characterization of the FRC included compressive and flexural tests, carried out according to NP EN 12390-3 (2011) [27]/NP EN 12390-13 (2013) [28] (compressive strength/elastic modulus) and EN 14,651 (2005) [29], respectively. The compressive tests were carried out using a minimum of 4 cylindrical specimens (diameter: 150 mm; height: 300 mm) per FRC batch. The FRC flexural properties were assessed using 4 prismatic specimens (width: 150 mm; height: 150 mm; length: 600 mm). Table 2 presents the main results in terms of mean values obtained from the material characterization, namely, the elastic modulus, $E_{\mathrm{cm}}$ and the compressive strength, $f_{\mathrm{cm}}$, from the compressive tests, and, from the flexural tests, the stress at limit of proportionality, $f_{\mathrm{ctl},L}$, calculated for a deflection $\delta_{L}$ = 0.05 mm, the equivalent flexural tensile strength $f_{\mathrm{eq},2}$ and $f_{\mathrm{eq},3}$, and the residual flexural tensile strength $f_{R1}$, $f_{R2}$, $f_{R3}$, and $f_{R4}$ for the crack mouth opening displacement (CMOD) of 0.5, 1.5, 2.5, and 3.5 mm, respectively. The FRC presents an average compressive strength of 46.4 MPa and an average elastic modulus of 25.7 GPa, which correspond to the concrete strength class C30/37, according to Eurocode 2 [30].

The GFRP’s top and bottom face sheets and webs were manufactured using the pultrusion process, and the corresponding mechanical properties were determined. The tensile properties (EN ISO 527 [31]) presented in Table 3 were obtained using a minimum of 6 specimens (thickness: 6 mm; width: 25 mm; length: 250 mm). The compressive and in-plane shear properties (ASTM D6641 [32] and ASTM D5379 [33]), assessed by [34], are presented in Table 4 and Table 5, respectively. The mechanical characterization included the evaluation of the tensile and compressive properties in the longitudinal and transverse directions of the pultruded GFRP profile.

Table 3 presents the longitudinal ($\sigma_{\mathrm{tu},L}$) and transverse ($\sigma_{\mathrm{tu},T}$) tensile strength, the longitudinal ($\varepsilon_{\mathrm{tu},L}$) and transverse ($\varepsilon_{\mathrm{tu},T}$) tensile strain at failure, and the longitudinal ($E_{\mathrm{tu},L}$) and transverse ($E_{\mathrm{tu},T}$) elastic modulus.

Table 3 presents the longitudinal ($\sigma_{\mathrm{cu},T}$) and transverse ($\sigma_{\mathrm{cu},L}$) compressive strength, the longitudinal ($\varepsilon_{\mathrm{cu},L}$) and transverse ($\varepsilon_{\mathrm{cu},T}$) compressive strain at failure, and the longitudinal ($E_{\mathrm{cu},L}$) and transverse ($E_{\mathrm{cu},T}$) elastic modulus. The results show a longitudinal and transverse tensile strength of ~304 MPa and 28 MPa, respectively, and a longitudinal and transverse elastic modulus of ~33 GPa and 5 GPa, respectively.

Table 5 presents the in-plane shear strength ($\tau_{u}$), the in-plane shear strain at failure ($\gamma_{u}$), and the shear modulus (G).

The core of the sandwich panel is made of a polyurethane (PUR) closed-cell foam with a density of 60 kg/m^3^.

During the production of the hybrid GFRP–FRC sandwich panel, an epoxy adhesive was used to promote the wet-bonded connection between the GFRP panel and the fresh wet layer of the FRC. Therefore, the adhesive connection between the GFRP plates and the FRC was previously studied, where 120 pull-out tests were carried out to evaluate the effects of (i) different types of adhesives, (ii) surface treatment methods, and (iii) durability on the performance of the adhesively bonded connection. Based on the results obtained, the commercial adhesive trademarked as “Sikadur-32 EF” was selected to promote the GFRP–FRC bond [35]. The commercial adhesive trademarked as “Sikadur-32 EF” was selected based on its performance to promote the GFRP–FRC bond [35]. According to the technical datasheet, with a density of 1500 kg/m^3^, this adhesive is a two-part bonding agent (2:1 component ratio, by weight or volume), with a pot life at 20 °C of 45 min and a viscosity of 1500 MPa·s.

### 2.2. Hybrid GFRP–FRC Sandwich Panels

In the development of the hybrid GFRP–FRC sandwich panel, genetic algorithms (GAs) were adopted to achieve a viable light-weight [36] solution with a lower carbon footprint and reduced manufacturing costs. GAs were defined in order to devise the best solution considering several boundary conditions: (i) the panel’s width had to be smaller than 500 mm (manufacturer’s requirement); (ii) the panel’s length was set to 5 m, as it satisfies the currents needs in the rehabilitation market; (iii) structural standards needed to be met, namely verification of the ultimate and serviceability limit states, throughout the EN 1990:2002 [37], the EN 1991-1-1:2002 [38], and the CNR DT 205/2007 [39]; and (iv) the acoustic and thermal performance needed to be evaluated. In addition to the former conditions, the panel’s design included the incorporation of a snap-fit type of connection between the panels. The final design is depicted in Figure 2a. Nevertheless, a simplified version of the final design was used in the development of the present work which does not present the snap-fit type of connection. The tested solution, presented in Figure 2b, has a width of 300 mm and a height of 160 mm and comprises a top layer made of FRC with a thickness of 20 mm. The panel’s top and face sheets and lateral webs are made of a 6-millimetre-thick GFRP profile and the panel’s core is made of polyurethane (PUR) foam (130 mm by 290 mm) with a density of 60 kg/m^3^.

The production of the sandwich panels included two main stages: (i) pultrusion of the composite sandwich (GFRP + PUR) and (ii) FRC casting. The pultrusion of the composite took place at ALTO—Perfis Pultrudidos, Lda. [40]. During this stage the core PUR foam blocks with the final dimensions (130 × 290 × 2000 (mm)) were introduced simultaneously with the unidirectional glass-fibre roving strands and fabrics in the heated die. An unsaturated polyester resin was used as the matrix of the GFRP and to promote the bond between the PUR core and the GFRP component (see Figure 3b). The architecture of the GFRP laminate is schematically presented in Figure 3a; it includes three layers of E glass fibre Roving (Roving 9600), four layers of woven roving combat (Combinate 450/800—woven fabric (density of 828 g/m^2^) combined with chopped strand mats (density of 450 g/m^2^)), and two layers of chopped strand mats (CSM). The fibre volume fraction of the GFRP laminate was approximately 51%.

The FRC casting was conducted at Civitest—Pesquisa de Novos Materiais para a Engenharia Civil, Lda. [41]. However, before pouring the fresh FRC, the top face sheet of the GFRP panel was slightly sanded and an epoxy adhesive (Sika 32 EF) was applied, in order to improve the bond between the former and the latter materials (see Figure 3c).

### 2.3. Experimental Programme

The experimental programme included three types of tests: (i) flexural tests with variable span (VST); (ii) flexural tests up to failure (FFT); and (iii) flexural creep tests (FCT). A total of 4 hybrid sandwich panels were tested, each one with a length of 4.7 m and a 300 mm by 160 mm cross-section (see Figure 2b). These four panels are presented in Table 6, alongside FRC casting date and FRC age upon testing.

#### 2.3.1. Flexural Tests with Variable Span

Static flexural tests with variable span were conducted to assess the global flexural stiffness throughout a three-point bending configuration. These tests were performed on two specimens (P1_F and P2_F) using the following three span lengths (L): 3.7 m, 4.0 m, and 4.5 m. Figure 4 shows the specimen’s geometry, test set-up, and instrumentations. A photo of the setup used for the flexural test with variable span is presented in Figure 5a. The load was applied at the midspan, and the two supports were equally distanced from the load point, by L/2. Both supports allowed for rotation, but only one allowed for longitudinal sliding. Metallic plates with a width of 50 mm and a thickness of 15 mm were placed between the supports and the hybrid panel. Additionally, the surface was evened out using a thin layer of plaster, thus ensuring perfect contact between the hybrid panels and the supports.

The variable span tests were composed of three cycles of loading and unloading force-controlled with a rate of 10 kN/min. Each cycle was composed of a loading stage up to 15 kN, followed by a 20 s plateau and a subsequent unloading stage to 0.1 kN. The maximum applied load of 15 kN corresponded to 30% of the maximum bending moment of the sandwich panel cross-section. This procedure was repeated for the three span lengths for each specimen. 

The instrumentation included linear variable differential transducers (LVDTs) and one load cell. Three LVDTs were used to record the vertical deformation of the panel at the midspan: LVDT 3 at the centre, and LVDT 2 and LVDT4 near the edges; see Figure 4b. All LVDTs had a linearity error of ±0.10%, and a range of ±50 mm (LVDT 2 and LVDT 4) or ±25 mm (LVDT 3). The load cell used to measure the applied load (F) had a maximum measuring capacity of 200 kN and a linear error of ±0.05%.

#### 2.3.2. Flexural Tests up to Failure

Once the flexural tests with variable span were concluded, the two hybrid panels P1_F and P2_F were monotonically tested in flexure up to failure according to a four-point bending test configuration. The specimens’ geometry, test set-up, and instrumentation are presented in Figure 6, whereas in Figure 5b a photo of the flexural test up to failure is shown. The load points, equally spaced from the mid-=span, were distanced by 1.5 m and the supports were placed at 100 mm from the panel’s ends. The instrumentation included 5 LVDTs, 4 strain gauges, and a load cell (see Figure 6), and was carried out under displacement control at a rate of 14 mm/min. The LVDT 2, LVDT 3, and LVDT 4 were placed at the midspan in the same position as the one described for the flexural tests with variable span (see Section 2.3.1). The LVDT 1 and LVDT 5 were placed at 750 mm from the midspan. All LVDTs had a linearity error of ±0.10% and a range of ±50 mm (LVDT 2 and LVDT 4) or ±25 mm (LVDT 1, LVDT 3, and LVDT 5). It should be noted that the LVDTs used had a ±50 mm range; therefore, in the ultimate stages of the flexural tests (when the midspan deformation was higher than 100 mm) the deflection was only measured with the actuator transducer. Two types of strain gauge types were used: (i) two TML PFL-30-11-3L strain sensors (SG3 and SG4) for measuring the midspan strain in the top layer of the FRC, and (ii) two TML BFLA-5-3 strain sensors (SG1 and SG2) for measuring the midspan strain in the bottom surface (GFRP laminate).

#### 2.3.3. Flexural Creep Tests

Flexural creep tests were conducted on two hybrid sandwich panels (P3_C and P4_C). Figure 7 presents the test set-up and instrumentation for the flexural creep tests. Each one was supported at two points and subjected to a distributed sustained load of 16.1 kN (11.93 kN/m) for a period of 2203 h (approximately 92 days). The long-term assessment of the creep behaviour included monitoring the panels after removing the creep load for a period of 2117 h (approximately 88 days).

The vertical deflection, strain in the top and bottom surfaces of the panel, temperature, and humidity were monitored during the long-term tests. The vertical deflection was obtained using three LVDTs (LVDT1 and LVDT2 at the panels’ extremities and LVDT2 at the panels’ midspan; see Figure 7a) and two mechanical dial gauges (DG1 and DG2, located at the midspan; see Figure 7b for each panel). The LVDTs had a range of ±25 mm and a linearity error of ±0.10%, whereas the mechanical dial gauges had a range of 40 mm and a graduation value of 0.01 mm. Similar to the flexural tests up to failure, four strain gauges were used to record the strain in the top layer of the FRC (SG3 and SG4; see Figure 7b) and the strain on the bottom surface of the GFRP (SG1 and SG2; see Figure 7b). The NI SCXI© system was used to record the displacements and strains, with an acquisition frequency of 1 Hz. The temperature and relative humidity (RH) were monitored with a digital thermohydrometre (EL-USB-2 EasyLog USB Data Logger with a range of −35 to +80 °C for temperature and 0 to 100% for RH).

The creep load was selected based on the results of the flexural failure tests with the aim of creating a bending moment equal to 20% of the ultimate bending moment. The gravity load was materialized using 25 kg cement bags, distributed along the panel, over 4.2 m. Loading was conducted as quickly as possible (within approximately 3 min) to minimize the occurrence of creep effects during the loading phase. Nevertheless, the gravity load was carefully positioned on the hybrid panel to minimize any dynamic loading effects. The same approach was used for unloading the panels. Figure 8 shows the hybrid sandwich panels before and after the application of the creep load.

## 3. Experimental Results

### 3.1. Flexural Test with Variable Span

The relationship between the force and the midspan displacement obtained from the flexural tests with variable span on panel P1_F is presented in Figure 9. For the load levels of these tests, the hybrid panels presented an almost linear elastic behaviour, showing great levels of recovery of the midspan deformation (~90%). The flexural stiffness, D, and the shear stiffness, U, can be computed based on the experimental results.

According to Timoshenko beam theory, the midspan deflection (δ) of a simply supported beam in a three-point bending configuration can be computed using Equation (1), where F is the applied load and L is the span length.
(1)$$\delta =\frac{F\cdot L^{3}}{48\cdot D}+\frac{F\cdot L}{4\cdot U}$$

Then, Equation (2) can be obtained by dividing Equation (1) by the term ·L:(2)$$\frac{\delta }{F\cdot L}=\frac{L^{2}}{48\cdot D}+\frac{1}{4\cdot U}$$

The flexural and shear stiffnesses of each panel can be obtained by means of a linear regression to the plot δ/(F·L) versus $L^{2}$, as depicted in Figure 10. Note that the linear regression is computed for each panel, considering the three different span tests, which corresponds to nine points (for each span *L*, the maximum force, 15 kN, was reached three times). Then, the slope of that regression, 1/(48·D), and the intercept with the vertical axis, 1/(4·U), can be used to estimate the flexural and shear stiffness values, respectively.

The results obtained are presented in Table 7. On average, the flexural (D) and shear (U) stiffnesses were equal to 1050.01 kN·m^2^ and 8772.73 kN, respectively. The flexural and shear stiffnesses were also computed analytically using Equations (3) and (4), respectively.
(3)D=E·I
(4)$$U=G\cdot A^{\prime }$$
where, E is the elastic modulus, I is the first moment of inertia, G is the shear modulus, and $A^{\prime }$ is the shear area. Using the results of material characterization described in Section 2.1, a D of 951.5 kN·m^2^ and a U of 5760.0 kN were obtained. Thus, there was good agreement between the experimental and analytical predictions, especially with the flexural stiffness, where the average experimental value was 9.4% and higher than the analytical D.

### 3.2. Flexural Tests up to the Failure

Table 8 presents the main results of the flexural tests up to failure, namely, the effective flexural stiffness ($K_{\mathrm{eff},\exp}$), ultimate load ($F_{\max}$), midspan deflection ($\delta_{\max}$) for $F_{\max}$, and maximum strain in the top layer of the FRC ($\varepsilon_{\mathrm{fu},\mathrm{FRC}}$). The effective flexural stiffness was computed between 5 kN and 15 kN of the applied load. The relationships between the applied force, F, and the midspan deflection are presented in Figure 11a, whereas Figure 11b presents the relationships between the F and the midspan strain in the top FRC layer. Figure 11b also presents the midspan strain values in the top (FRC) and bottom (GFRP) surfaces of the panel according to a cross-section analytical prediction.

Both specimens (P1_F and P2_F) showed identical force versus midspan deflection responses, with an effective flexural stiffness of 0.65 kN/mm. In general, the hybrid panels presented an almost linear behaviour up to failure. However, a slight stiffness reduction can be seen for midspan displacements higher than 60 mm. 

Through an analytical approach (i.e., a cross-sectional analysis considering strain compatibility and conventional force equilibrium in the cross-section), in which it was assumed that the materials composing the hybrid panel displayed linear behaviour, the strain in the FRC upper layer ($\varepsilon_{\mathrm{fu},\mathrm{FRC}}$) obtained experimentally at failure (defined by the ultimate moment: 53.36 kN·m) was validated (difference of 2.1%). As can be seen in Table 8, on average the $\varepsilon_{\mathrm{fu},\mathrm{FRC}}$ was equal to 0.26%, which was smaller than the ultimate FRC strain in compression (~0.350%). Due to technical problems, the data from the strain gauges placed at the bottom GFRP layer had to be disregarded. However, based on the analytical approach, the strain in the bottom layer at failure would be around 0.57% (see Figure 11b).

Failure was observed when the applied force reached 71 kN and the vertical displacement at the midspan was, on average, 129 mm. Both specimens showed local web buckling due to transverse compression at the point loads, which triggered the failure, followed by FRC crushing. Figure 12 presents the failure modes observed in both panels. It should be noted that, despite the FRC crushing, debonding between the FRC layer and the GFRP profile was not observed.

### 3.3. Creep Flexural Tests

Figure 13 presents the evolution of midspan deflection over time for the two hybrid panels subjected to the creep load, as well as the temperature variation. Due to technical problems, it was not possible to monitor the temperature throughout the entire test. It should be noted that, despite the test being conducted indoors, at the beginning of the creep tests the average air temperature was 26 °C (summer season), whereas after 5 months, at the end of the monitoring, the average air temperature was close to 10 °C (winter season). The instantaneous vertical displacements measured after the application of the gravity loading ($\delta_{e,\mathrm{lo}}$), the displacement due to the creep effect ($\delta_{\mathrm{cr},\mathrm{lo}}$), the instantaneous upwards displacement due to the removal of the gravity load ($\delta_{e,\mathrm{un}}$), and the recovered midspan displacement over the 88 days ($\delta_{\mathrm{cr},\mathrm{un}}$) are presented in Table 9. The instantaneous vertical displacement at the midspan upon loading (downwards: 22.21 mm, CoV = 5%) was similar to the instantaneous vertical displacement at the midspan when the load was removed (upwards: 21.67 mm, CoV = 4%). Therefore, these results indicate that the flexural behaviour (namely its flexural stiffness) was not significantly affected by the creep loading.

During the 2203 testing hours, an average increase in the midspan displacement of about 5.85 mm was observed, which represents 20.8% of the total midspan displacement. This deformation, caused by the viscoelasticity of the materials that constitute the panel, corresponded to a creep coefficient of 0.27. The creep coefficient $\varphi_{\mathrm{lo}}$ was computed based on the following Equation (5):(5)$$\varphi_{\mathrm{lo}}=\frac{\delta_{\mathrm{cr},\mathrm{lo}}}{\left(\delta_{e,\mathrm{lo}}+\delta_{\mathrm{cr},\mathrm{lo}}\right)}$$

During the recovery stage (after unloading the panels), the midspan deformation was also monitored and, after 2117 h, an upwards displacement of 2.95 mm at the midspan was observed. Based on these results, the creep recovery coefficient $\varphi_{\mathrm{un}}$ was computed based on Equation (6):(6)$$\varphi_{\mathrm{un}}=\frac{\delta_{\mathrm{cr},\mathrm{un}}}{\left(\delta_{e,\mathrm{un}}+\delta_{\mathrm{cr},\mathrm{un}}\right)}$$

The obtained creep coefficient and creep recovery coefficient are presented in Table 9. When compared with $\varphi_{\mathrm{lo}}$, the creep recovery coefficient was smaller ($\varphi_{\mathrm{un}}=0.14$). However, it should be noted that the air temperature at the beginning of this long-term test (during the creep loading) was higher than at the end (during the recovery stage), and this might have interfered with the viscoelastic response of the composing materials.

Figure 14 presents the long-term monitoring of strain at the midspan on the top FRC layer (SG3 and SG4) and the bottom surface of the GFRP (SG1 and SG2) for both hybrid panels. It should be noted that the strain values registered immediately after the loading agree with the analytical predictions (cross-section analysis). The values from the analytical predictions are also presented in Figure 14 with a red and blue dashed line for the strain values of the top FRC layer and the bottom surface of the GFRP, respectively. When the gravity load was removed, the strains in the top (FRC) and bottom (GFRP) surfaces of the panel were similar. However, the values of these strains were not equal to zero and they presented a continuous increase overtime. This observation can be related to the temperature variation, which was about 16 °C lower at the end of the test.

Finally, it should be mentioned that during the creep tests no slip at the FRC–GFRP interface was observed.

## 4. Analytical Modelling

The flexural creep response of the GFRP profiles was already obtained using Findley’s power law [42] in the literature (e.g., [43,44,45,46,47]). Recently, Gonilha et al. [20] proposed a composed creep model (CCM) based on the creep response of the constituent materials to predict the long-term response of GFRP–concrete hybrid structures. Therefore, in addition to Findley’s power law, the CCM was also considered in this study for the analytical modelling of hybrid GFRP–FRC sandwich panels.

### 4.1. Findley’s Power Law

In order to study the viscoelastic behaviour of hybrid GFRP–FRC sandwich panels, Findley’s power law was applied in this study. Findley’s power law is commonly used to predict the creep response of GFRP materials. Equation (7) shows the formulation of Findley’s power law.
(7)$$\Delta \left(t\right)=\Delta_{0}+m\cdot t^{n}$$
where Δ(t) is the time-dependent deformation, $\Delta_{0}$ is the instantaneous deformation, t represents time, m is a creep amplitude coefficient depending on the applied stress, and n is a time exponent coefficient, independent of stress. The coefficient n is assumed to be a material dependent on a given hygrothermal condition [48]. Equation (7) can be rearranged and written as follows:(8)$$\log\left(\Delta \left(t\right)-\Delta_{0}\right)=\log m+n\cdot \log t$$

Equation (8) gives a straight line when plotted on a log/log scale with the intercept equal to m and the slope equal to n. For the unloading stage, Findley’s power law can be useful as well. Figueira et al. [49], considering Findlay’s power law, predicted the time-dependent deformation at the unloading stage:(9)$$\Delta \left(t\right)=m\cdot {\left(t_{un}\right)}^{n}+m_{un}\cdot {\left(t-t_{un}\right)}^{n}$$
where $t_{un}$ is the age at which unloading is conducted and $m_{un}$ is the creep amplitude coefficient at the unloading stage. According to Figueira et al., $m_{un}$ must assume a negative value, and a smaller absolute value compared to the loading stage. There is no need to change the time exponent coefficient *n* for the unloading stage, and the value considered in the loading stage can be used [49].

The creep deformation obtained per panel is illustrated in Figure 15 in terms of creep deformation at the loading stage (Figure 15a) and total deformation at both stages (Figure 15b). Fitting the power law to the experimental data, the coefficients m and n were obtained with high coefficients of determination $\left(R^{2}\right)$ of 0.936 and 0.943 for P3_C and P4_C, respectively. According to Figure 15a, at the first 10 h of testing, the experimental values exhibited a different pattern of variation from the model values registered. In fact, the experimental values increased with a higher rate and afterward progressively with a lower rate characterized by the n value obtained from the power law fitting. As may be observed in Figure 15b, satisfactory fitting can be obtained between experimental and modelling values. Especially for the loading stage, which presents less variability compared to the unloading stage. The resulting power law parameters, including the coefficients m and n and also the instantaneous deformation $\left(\Delta_{0}\right)$, are presented in Table 10.

### 4.2. Composed Creep Model

A composed creep model (CCM) is a model which is used to simulate the sandwich panel’s creep deflections by considering the individual viscoelastic contributions, such as GFRP faces, webs, and FRC layer. According to the other studies [20,22,50], it is expected that viscoelastic shear deformation reach relevant proportions in the total deformation of GFRP structures subjected to flexure. Therefore, in this case, the Timoshenko beam theory was used to estimate the total deflection (flexural and shear deflection). In the following sections, the instantaneous and time-dependent deflections of the sandwich panels under study are estimated according to Timoshenko beam theory.

#### 4.2.1. Instantaneous Deflection

Based on Timoshenko beam theory, for the load model presented in Figure 16, the instantaneous deflection at the midspan may be calculated by Equation (10):
(10)$$\delta_{mid-span}=\frac{C_{1}}{EI}+\frac{C_{2}}{G\cdot \left(k\cdot A\right)}$$
where the first term represents the flexure deflection and the second term the shear deflection; the terms $C_{1}$ and $C_{2}$ are given by Equations (11) and (12). For the shear stiffness, calculations were performed assuming that it was provided only by the GFRP webs, as suggested in [22]; therefore, the general shear area (k·A) may be substituted by the area of the webs $(A_{r}$), while the general shear modulus (G) should be substituted by the in-plane shear modulus of the webs $\left(G_{LT\left(r\right)}\right)$. EI is the equivalent section flexural stiffness, calculated by Equation (13), for which two types of elemental areas were considered: (i) FRC layer and (ii) GFRP faces.
(11)$$C_{1}=q\times \left[\frac{a^{3}b}{6}+\frac{a^{2}b^{2}}{4}+\frac{5ab^{3}}{48}+\frac{5b^{4}}{384}\right]$$
(12)$$C_{2}=q\times \left[\frac{ab}{2}+\frac{b^{2}}{8}\right]$$
(13)$$EI=\overset{n}{\underset{i}{\sum }}E_{i}I_{i}+E_{i}A_{i}\cdot {\left(NA-z_{gi}\right)}^{2}$$
where $E_{i}$ is the elasticity modulus of element *i,* $A_{i}$ is the area of element *i*, and $I_{i}$ is the second moment of area of each element i around its own stiffness centroid ($z_{gi}$). Furthermore, accounting for the orthotropic nature of the GFRP material, the elasticity modulus in the longitudinal direction ($E_{L,i}$) of the GFRP faces should be used. NA is the position of the cross-section neutral axis, which may be determined by Equation (14):(14)$$NA=\frac{\sum_{i}^{n}E_{i}A_{i}z_{i}}{\sum_{i}^{n}E_{i}A_{i}}$$
where $z_{i}$ is the distance between the stiffness centroid of element *i* and a chosen specified axis. These formulae allowed for the determination of the shear and flexural stiffness of the cross-section G·(k·A)=6048 kN and $EI=972\;\mathrm{kN}\cdot m^{2}$, respectively. Table 11 and Table 12 show the parameters used to determine the shear and flexural stiffness of the cross-section. 

The instantaneous deflection at the midspan of the hybrid GFRP–FRC sandwich panels predicted by Timoshenko beam theory, presented in Table 13, compares well with the results observed experimentally.

#### 4.2.2. Time-Dependent Deflection

To determine the long-term deflection of hybrid GFRP–FRC sandwich panels, the elastic moduli (E and G) introduced in Equation (10) should be replaced by time-dependent moduli using the models based on the creep response of the constituent materials.

One of the major concerns associated with the use of GFRP pultruded material in construction is their susceptibility to creep effects [51,52]. Such concerns are also driven by the limited age of application of these materials—usually, a service life of at least 50 years is required for most civil engineering structures. The viscoelastic behaviour of these materials must be considered in the analysis and design of any structure. Bank [22], based on Findley’s power law, proposed general creep models for GFRP pultruded profiles in service conditions subjected to flexure:(15)$$E\left(t\right)=\frac{1241.1\cdot E_{0}\;}{1241.1+E_{0}\cdot t^{0.3}}$$
(16)$$G\left(t\right)=\frac{186.2\;\cdot G_{0}\;}{186.2+G_{0}\cdot t^{0.3}}$$
where E(t) and G(t) are the time-dependent moduli, $E_{0}$ and $G_{0}$ are the instantaneous moduli in gigapascal, and t is the time in hours. This theory has been applied to determine the creep constants for conventional pultruded sections by a number of authors.

The EuroComp design code and handbook [25] suggests a time-dependent creep reduction factor curve for time-dependent moduli of unidirectional GFRP composites in tension and shear, given by the following equations:(17)$$E\left(t\right)=E_{0}\left(0.992-5.965\times 10^{-3}\cdot lnt\right)$$
(18)$$G\left(t\right)=G_{0}\left(0.897-4.719\times 10^{-2}\cdot lnt\right)$$

Equations (17) and (18), with the time in hours, are valid for t≥0.1 h. Unlike the shear creep model presented by Bank [22], the model presented in the EuroComp design code and handbook [25] does not specifically refer to shear in flexure but only shear loading. 

In order to consider the effect of temperature on the creep rates of the GFRP material by the models presented herein, Gonilha et al. [20] proposed Equations (19)–(22), respectively, for the case of the expressions proposed by Bank and EuroComp:(19)$$E\left(t\right)=\frac{1241.1\;E_{0}\;}{1241.1+E_{0}\cdot t^{0.3\left(T/T_{0}\right)}}$$
(20)$$G\left(t\right)=\frac{186.2\;G_{0}\;}{186.2+G_{0}\cdot t^{0.3\left(T/T_{0}\right)}}$$
(21)$$E\left(t\right)=E_{0}\left(0.992-\left(T/T_{0}\right)\;5.965\times 10^{-3}\cdot lnt\right)$$
(22)$$G\left(t\right)=G_{0}\left(0.897-\left(T/T_{0}\right)\;4.719\times 10^{-2}\cdot lnt\right)$$
where $T_{0}$ is the reference temperature for which the regression parameters were determined and T is the temperature for which the long-term creep behaviour was being predicted (in °C). 

In this study, the creep behaviour of GFRP material was estimated according to the recommendations of Gonilha et al. [20]. As suggested in [20], the creep models should be chosen based on the actual stress distribution, i.e., considering the type of stress each material is subjected to. Based on the neutral axis in the sandwich panel (NA=115.3 mm, with respect to the lowest fibre of the cross-section), the FRC layer is in compression while the GFRP profile is mainly in tension. Therefore, in this study a tension creep model (similar to the one proposed by the EuroComp design code and handbook [25]) was used to determine the time-dependent GFRP elasticity modulus, instead of a flexure creep model (such as the one proposed by Bank [22]). In addition, for the time-dependent shear modulus, a shear in the flexure creep model (such as the one proposed by Bank [22]) was used since it is assumed to be more appropriate than a pure shear creep model (such as the one proposed in the EuroComp design code and handbook [25]). 

Concrete structures are also susceptible to creep phenomena [23]. To estimate the long-term deflection due to creep, the creep behaviour of the FRC layer was estimated according to the recommendations of Eurocode 2-Annex B [23]. Therefore, based on the concrete creep law, to estimate the time-dependent concrete modulus of elasticity, Equation (23) was used: (23)$$E_{c}\left(t,t_{0}\right)=\frac{E_{c,28}}{1+\chi \left(t,t_{0}\right)\cdot \varphi \left(t,t_{0}\right)}$$
where $E_{c}\left(t,t_{0}\right)$ is the time-dependent concrete elasticity modulus, $E_{c,28}$ is the concrete elasticity modulus at 28 days, $\chi \left(t,t_{0}\right)$ is Trevino’s ageing coefficient given by Equation (24) [20], and $\varphi \left(t,t_{0}\right)$ is the concrete creep coefficient, which is determined by Equation (25):(24)$$\chi \left(t,t_{0}\right)≅\chi \left(t_{0}\right)=\frac{\sqrt[{3}]{t_{0}}}{1+\sqrt[{3}]{t_{0}}}$$
(25)$$\varphi \left(t,t_{0}\right)=\varphi_{0}\cdot \beta_{c}\left(t-t_{0}\right)$$
where t is the time in days and $t_{0}$ is the age of the concrete at the time of loading, in days. $\varphi_{0}$ is the notional creep coefficient and $\beta_{c}\left(t,t_{0}\right)$ is the coefficient for describing the development of creep with time after loading, the details of which are discussed in Appendix A. The environmental conditions of the creep tests were considered by introducing into the creep models an average temperature of T=21 °C and a relative humidity of RH=56%.

The comparison between the deformation predicted with the composed creep model and the experimental results is illustrated in Figure 17 in terms of creep deformation (Figure 17a) and total deformation (Figure 17b).

Figure 18 compares the long-term creep deformations predicted by the composed creep model with those predicted by Findley’s power law. Table 14 summarizes the comparison of results after 20, 50, and 100 years. 

As shown in Figure 18, the deformations predicted by Findley’s power law diverges considerably from those predicted by the composed creep model. According to Gonilha et al. [20], Findley’s power law is not adequate for predicting the long-term creep deformation of hybrid GFRP–FRC sandwich panels. They explained that the changes in the neutral axis of the section in creep flexure may change the logarithmic slope of the curve representing the time exponent coefficient (coefficient n), which is constant in Findley’s power law. Furthermore, the creep behaviour of concrete does not follow Findley’s power law. However, as suggested in [20], the composed creep model is able to predict the long-term creep deformation of hybrid GFRP–FRC sandwich panels by considering the important effects of environmental changes (temperature and relative humidity).

Regarding the limitations which have been considered for the long-term deflection of real structures, Eurocode 2 [30] limits the deflection after construction to L/500 (quasi-permanent load combination, with L being the span). In this case, considering the span of the panels under study (4500 mm), the deflection should be limited to 9 mm. Using the composed creep model described in this study, for a quasi-permanent load (Equation (26)), the hybrid panel is expected to present a midspan deflection of 5.98 mm (125% of the instantaneous deflection) after 50 years.
(26)$$p=g_{k}+\psi_{2}\cdot q_{k}$$

In Equation (26), p is the service load (quasi-permanent combination), $g_{k}$ is the permanent load, which comprises the self-weight of the panel (0.6 $\mathrm{kN}/m^{2}$) and other permanent loads (1.5 $\mathrm{kN}/m^{2}$), $q_{k}$ is the variable load (2 $\mathrm{kN}/m^{2}$), and $\psi_{2}$ is the factor for the quasi-permanent value of a variable load (0.3).

The portions of flexural force taken by the FRC layer and GFRP faces may be estimated based on Equations (27) and (28). It is assumed that $\alpha_{FRC}+\alpha_{GFRP}=1$, i.e., the only panel components contributing to the flexural stiffness of the panel are the FRC layer and the GFRP faces, and these contributions change with creep time according to the time-dependent properties of the materials.
(27)$$\alpha_{FRC}\left(t\right)=\frac{{\left(EI\right)}_{FRC}}{{\left(EI\right)}_{FRC}+{\left(EI\right)}_{GFRP}}$$
(28)$$\alpha_{GFRP}\left(t\right)=\frac{{\left(EI\right)}_{GFRP}}{{\left(EI\right)}_{FRC}+{\left(EI\right)}_{GFRP}}$$

Figure 19 plots the time-dependent $\alpha_{FRC}$ and $\alpha_{GFRP}$ factors for a long period (100 years). The relative contribution of the GFRP faces to the panel’s flexural stiffness increases over time. This is because the flexural modulus is more significantly reduced in the FRC compared to the GFRP faces. This result indicates that the flexural load is partly transferred from the FRC to the GFRP faces over time.

## 5. Conclusions

This paper presented experimental and analytical investigations on the short- and long-term behaviour of hybrid GFRP–FRC sandwich panels. The developed hybrid panels have a rectangular cross-section of 300 mm × 160 mm (width × height), with a FRC top layer with a thickness of 20 mm and GFRP bottom and lateral web face sheets with thicknesses of 6 mm. The experimental programme included different types of tests: (i) flexural tests with variable span, (ii) flexural tests up to failure, and (iii) flexural creep tests. From the experimental programme, the following conclusions can be drawn:In the flexural tests up to failure, the developed hybrid panels presented an almost linear behaviour. The failure mode was characterized by local web buckling at the point loads, followed by FRC crushing.In the flexural creep tests, the instantaneous loading and unloading deflections were similar and approximately equal to the values registered in the flexural tests up to failure. Furthermore, after approximately 2000 h of loading, a creep coefficient of 0.27 was obtained.Finally, an adequate bond between the FRC and the GFRP was achieved in the tested panels, as no slip at the FRC–GFRP interface was observed.

In the analytical part of this study, the long-term response of hybrid GFRP–FRC panels was predicted by using Findley’s power law and a composed creep model (CCM) based on the creep response of the constituent materials. The analytical study allowed us to draw the following conclusions:The analytical studies showed that the long-term response of hybrid GFRP–FRC panels can be predicted by using the CCM based on the creep response of the constituent materials taking into account the environmental conditions (temperature and relative humidity).By using CCM, it is expected that the developed panel will comply with the provisions of Eurocode 2 [30] in terms of deflection for a service life of 50 years under a quasi-permanent load combination.Based on analytical modelling, the transfer of flexural load from the FRC to the GFRP is predicted to occur due to the higher flexural modulus reduction in the FRC, indicating that the FRC contribution to the flexural stiffness of the panels becomes reduced over time.

## Figures and Tables

**Figure 1 materials-15-02536-f001:**
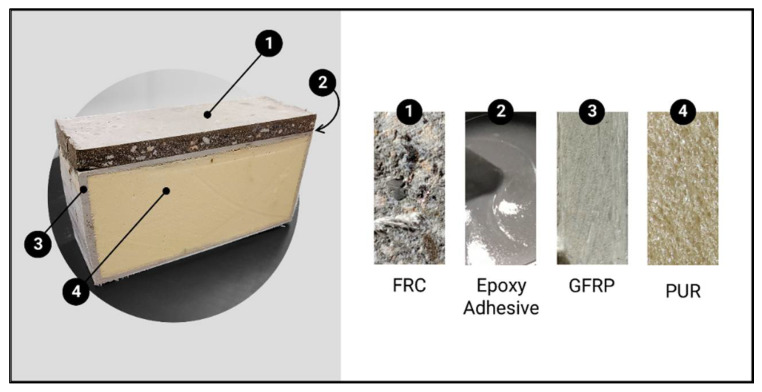
Photo of the hybrid sandwich panel and constituent materials.

**Figure 2 materials-15-02536-f002:**
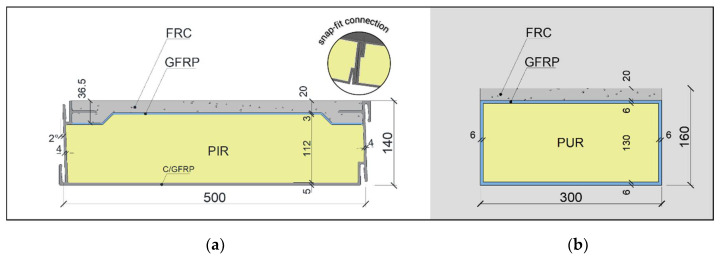
Final geometry of the hybrid sandwich panel: (**a**) final design and (**b**) tested solution. Units in mm.

**Figure 3 materials-15-02536-f003:**
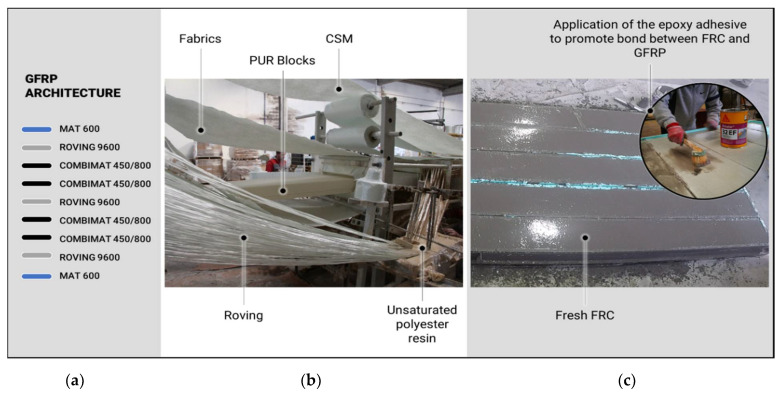
Production of the sandwich panels: (**a**) GFRP laminate architecture, (**b**) pultrusion process, and (**c**) FRC casting.

**Figure 4 materials-15-02536-f004:**
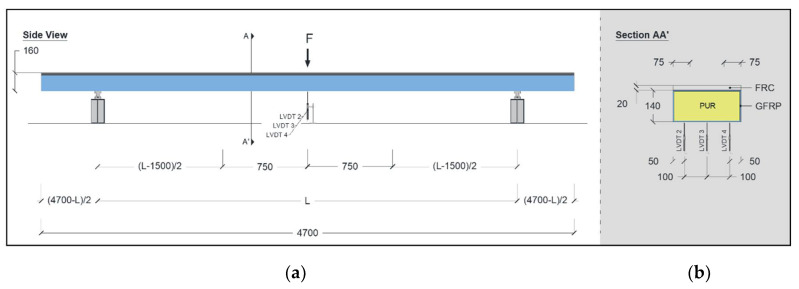
Geometry, test set-up, and instrumentation for the static test with variable span: (**a**) side view and (**b**) cross-section view. Units in mm.

**Figure 5 materials-15-02536-f005:**
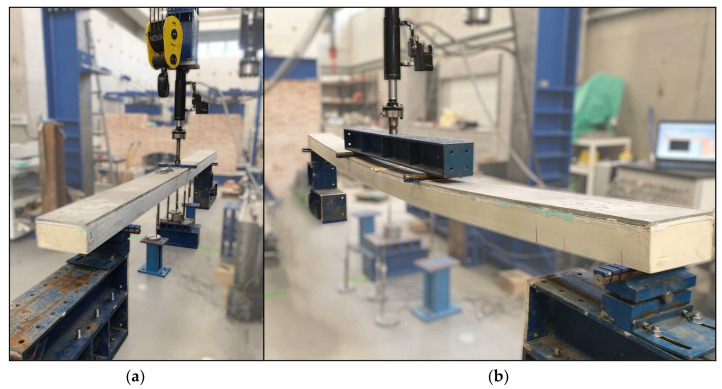
Photo of test set-up for (**a**) static flexural test with variable span and (**b**) flexural failure tests.

**Figure 6 materials-15-02536-f006:**
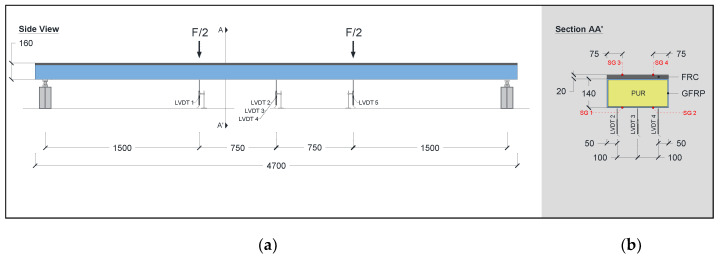
Geometry, test set-up, and instrumentation for the flexural tests up to failure: (**a**) side view and (**b**) cross-section view. Units in mm.

**Figure 7 materials-15-02536-f007:**
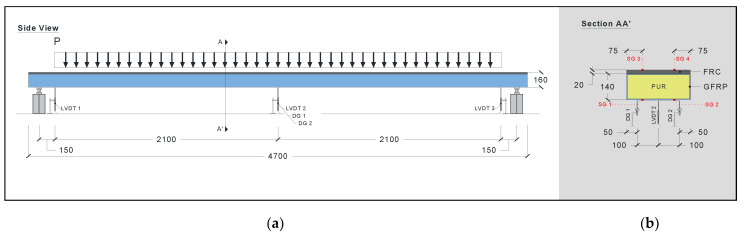
Geometry, test set-up and instrumentation for flexural creep tests: (**a**) side view and (**b**) cross-section view. Units in mm.

**Figure 8 materials-15-02536-f008:**
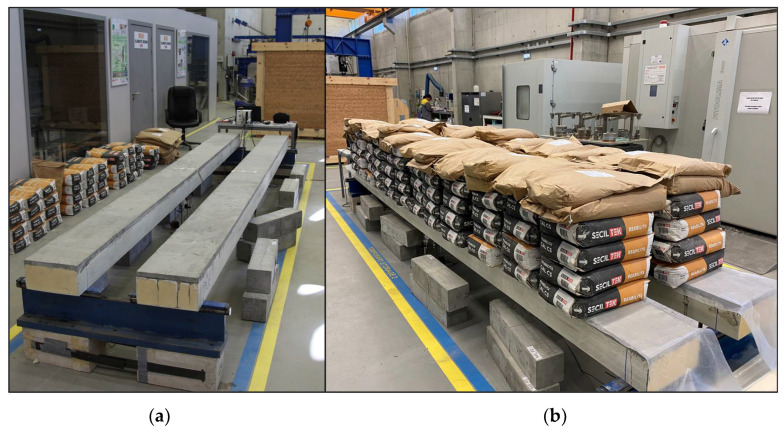
Photo of flexural creep tests: (**a**) before the application of the creep load and (**b**) with the creep load.

**Figure 9 materials-15-02536-f009:**
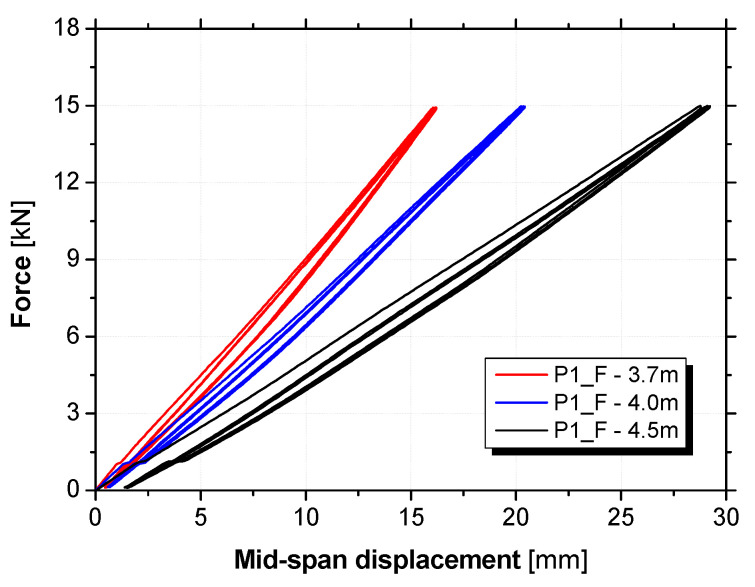
Force versus midspan displacement of the flexural tests with variable span (panel P1_F).

**Figure 10 materials-15-02536-f010:**
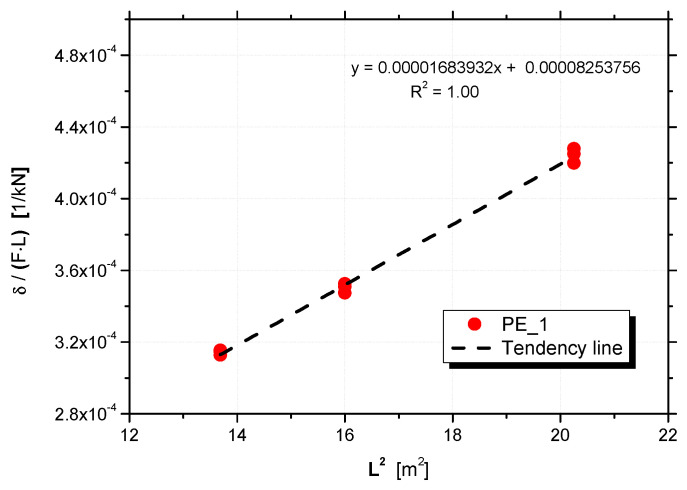
Linear regression of the plot δ/(F·L) versus $L^{2}$, for panel PE_1.

**Figure 11 materials-15-02536-f011:**
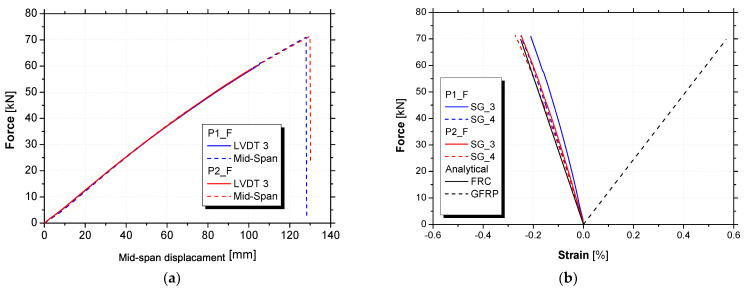
Flexural tests up to failure: (**a**) force versus midspan deflection and (**b**) force versus midspan strain in GFRP and FRC.

**Figure 12 materials-15-02536-f012:**
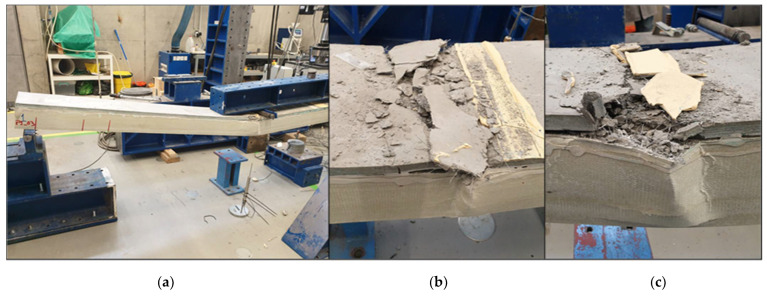
Photo of the failure mode—local bucking on the side webs of the GFRP profile and FRC crushing: (**a**) location of failure, (**b**) detail of failure in panel P1_F, and (**c**) detail of failure in panel P2_F.

**Figure 13 materials-15-02536-f013:**
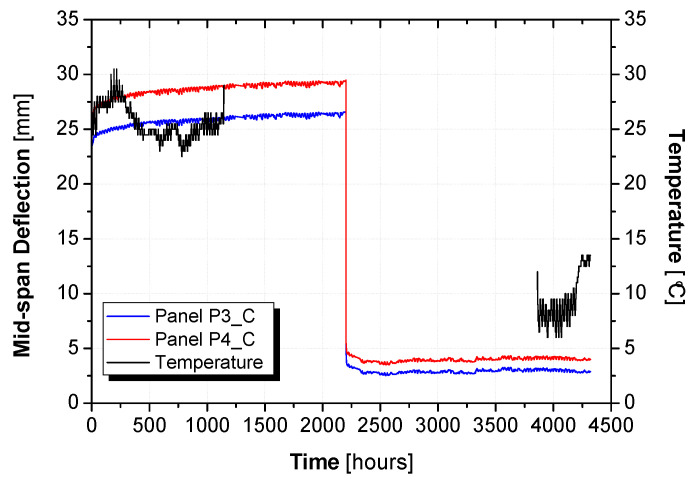
Mid-span displacement variation with time.

**Figure 14 materials-15-02536-f014:**
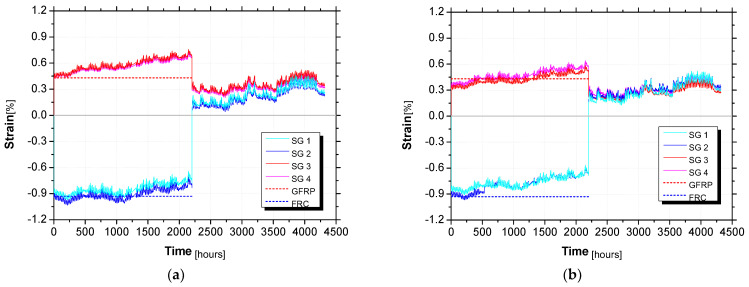
Strain variation for panel (**a**) P3_C and (**b**) P4_C.

**Figure 15 materials-15-02536-f015:**
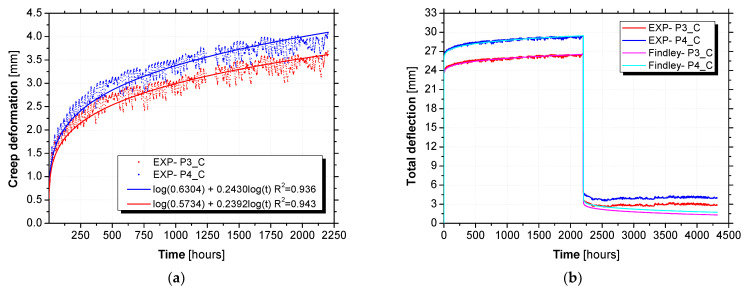
Findley’s power law values adjusted to the experimental values: (**a**) creep deformation at loading stage (log10 plot) and (**b**) total deformation at loading and unloading stages.

**Figure 16 materials-15-02536-f016:**
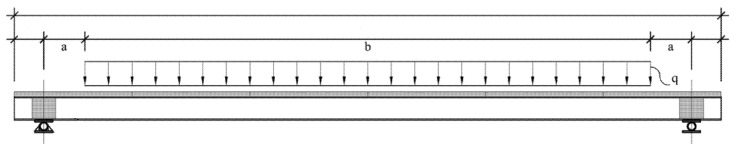
Load configuration of creep tests. Units in mm.

**Figure 17 materials-15-02536-f017:**
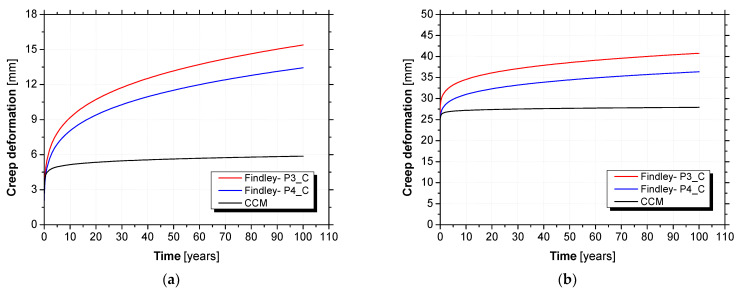
Composed creep model versus experimental results: (**a**) creep deformation and (**b**) total deformation.

**Figure 18 materials-15-02536-f018:**
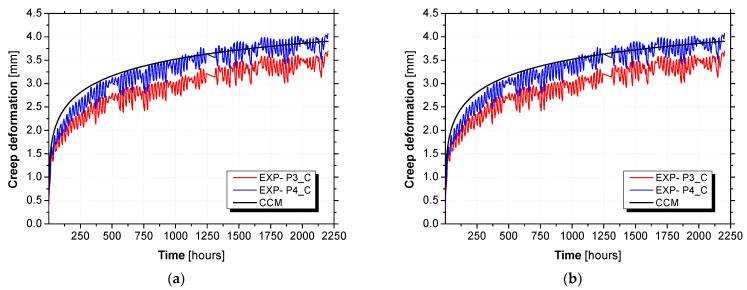
Long-term predictions of composed creep model versus Findley’s power law: (**a**) creep deformation and (**b**) total deformation.

**Figure 19 materials-15-02536-f019:**
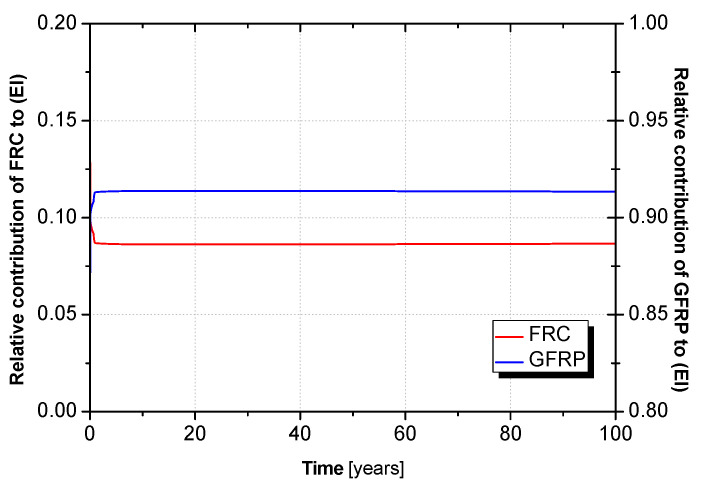
Time-dependent relative contributions of the GFRP faces ($\alpha_{GFRP}$) and FRC ($\alpha_{FRC}$) layer to the panel’s flexural stiffness (EI).

**Table 1 materials-15-02536-t001:** Composition of the FRC mixture.

Composition	Amount of Material per m^3^ of Mixture (kg)
Cement CEM I 42.5 R	376
Water	196
Fine sand	156
Gravel 12.5 mm	626
Fly ash	200
Superplasticizer	12
Steel fibres	60
Polypropylene	2

**Table 2 materials-15-02536-t002:** Properties of the FRC (mean values).

Batch	$f_{cm}$ (MPa)	$E_{cm}$ (GPa)	$f_{ctL}$ (MPa)	$f_{eq2}$ (MPa)	$f_{eq3}$ (MPa)	$f_{R1}$ (MPa)	$f_{R2}$ (MPa)	$f_{R3}$ (MPa)	$f_{R4}$ (MPa)
B1	49.61 (4.64%)	26.91 (6.87%)	6.13 (15.30%)	12.51 (9.47%)	11.78 (12.02%)	12.22 (9.27%)	12.07 (10.99%)	10.92 (10.42%)	9.37 (12.48%)
B2	43.26 (2.94%)	24.47 (4.66%)	3.54 (30.15%)	7.52 (27.98%)	7.18 (27.29%)	7.30 (31.51%)	7.60 (22.88%)	6.60 (26.69%)	5.69 (28.31%)

Note: The values between parentheses are the corresponding coefficients of variation (CoV).

**Table 3 materials-15-02536-t003:** Tensile properties of the GFRP face sheets/webs.

GFRP	$\sigma_{tu,L}$ (MPa)	$\varepsilon_{tu,L}$ (mm/m)	$E_{tu,L}$ (GPa)	$\sigma_{tu,T}\;$ (MPa)	$\varepsilon_{tu,T}$ (mm/m)	$E_{tu,T}$ (GPa)
Face sheets	344.49 (18.37%)	11.53 (18.59%)	33.55 (11.65%)	28.06 (28.97%)	8.62 (46.32%)	4.93 (7.42%)
Webs	263.49 (13.06%)	9.85 (3.77%)	31.51 (15.96%)	22.9 (29.66%)	-	-

Note: The values between parentheses are the corresponding coefficients of variation (CoV).

**Table 4 materials-15-02536-t004:** Compressive properties of GFRP face sheets/webs (adapted from [34]).

$\sigma_{cu,L}$ (MPa)	$\varepsilon_{cu,L}$ (mm/m)	$E_{cu,L}$ (GPa)	$\sigma_{cu,T}\;$ (MPa)	$\varepsilon_{cu,T}$ (mm/m)	$E_{cu,T}$ (GPa)
322.80 (15.71%)	12.33 (5.56%)	29.80 (11.41%)	93.5 (16.11%)	10.29 (20.28%)	10.30 (3.88%)

Note: The values between parentheses are the corresponding coefficients of variation (CoV).

**Table 5 materials-15-02536-t005:** In-plane shear properties of the GFRP webs (adapted from [34]).

$\tau_{u}$ (MPa)	$\gamma_{u}$ (mm/m)	*G* (GPa)
50.23 (2.81%)	15.00 (12.67%)	3.6 (7.80%)

Note: The values between parentheses are the corresponding coefficients of variation (CoV).

**Table 6 materials-15-02536-t006:** Experimental programme.

Panel	Casting Date	Batch	FRC Age at Testing Date (Days)		
			VST	FFT	FCT
P1_F	14/02/2020	B1	87	87	-
P2_F	14/02/2020	B1	90	90	-
P3_C	12/05/2020	B2	-	-	77
P4_C	12/05/2020	B2	-	-	77

**Table 7 materials-15-02536-t007:** Flexural and shear stiffness values obtained experimentally.

Panel	*D* (kN m)	*U* (kN)
P1_F	1038.7	10,314.8
P2_F	1061.2	7230.7
(Average)	1050.0	8772.8

**Table 8 materials-15-02536-t008:** Main results from the four-point bending tests.

Panel	$F_{u}$ (kN)	$d_{u}$ (mm)	$M_{u}$ (kN m)	$K_{eff}$ (kN/mm)	$\varepsilon_{fu,FRC}$ (%)	Failure Mode
P1_F	70.94	127.89	53.21	0.650	−0.249	Buckling of the webs followed by concrete crushing at the load point
P2_F	71.35	129.91	53.51	0.657	−0.272	Buckling of the webs followed by concrete crushing at the load point

**Table 9 materials-15-02536-t009:** Main results from the flexural creep tests.

Panel	$\delta_{e,lo}$ (mm)	$\delta_{cr,lo}$ (mm)	$\varphi_{lo}$ (-)	$\delta_{e,un}\;$ (mm)	$\delta_{cr,un}$ (mm)	$\varphi_{un}$ (-)
P3_C	21.12	5.51	0.26	20.82	2.93	0.14
P4_C	23.29	6.19	0.27	22.52	2.97	0.13

**Table 10 materials-15-02536-t010:** Coefficients m and n calibrated from the experimental tests.

Specimen	Applied Load (kN/m)	Loading Stage			Unloading Stage	
		$\Delta_{0}$ (mm)	*m* (mm)	*n* (-)	$m_{un}$ (mm)	n (-)
P3_C	3.76	22.93	0.5734	0.2392	0.3688	0.2392
P4_C	3.76	25.38	0.6304	0.2430	0.3600	0.2430

**Table 11 materials-15-02536-t011:** Parameters used to determine the flexural stiffness of the cross-section.

Elemental Areas	$E_{i}$ (GPa)	$A_{i}$ (mm^2^)	$I_{i}$ (mm^4^)
FRC layer	24.47	600	6.24 × 10^6^
GFRP layer	33.55	3600	24.56 × 10^6^

**Table 12 materials-15-02536-t012:** Parameters used to determine the shear stiffness of the cross-section.

Elemental Areas	$G_{LT\left(r\right)}$ (GPa)	$A_{r}$ (mm^2^)
GFRP webs	3.6	1680

**Table 13 materials-15-02536-t013:** Comparison between instantaneous deflection obtained from the experimental results and that obtained from analytical modelling.

Specimen	$\delta_{mid-span}$ (mm)
Experimental-P3_C	22.93
Experimental-P4_C	25.38
Analytical	22.05

**Table 14 materials-15-02536-t014:** Predictions of long-term creep deflections: composed creep model (CCM) and Findley’s power law.

Specimen		t=20 (Years)	t=50 (Years)	t=100 (Years)
Findley—P3_C	Δ (mm)	9.39	11.51	13.43
	$\Delta \delta /\delta_{0}$ (%)	40.95	50.20	58.57
Findley—P4_C	Δδ (mm)	10.72	13.17	15.39
	$\Delta \delta /\delta_{0}$ (%)	42.24	51.89	60.64
Composed creep model (CCM)	Δδ (mm)	5.35	5.63	5.87
	$\Delta \delta /\delta_{0}$ (%)	24.26	25.53	26.62

## Data Availability

Not applicable.

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
