# Peer review of "Flexural Creep Response of Hybrid GFRP–FRC Sandwich Panels"

_materials, 2022, doi:10.3390/ma15072536_

Round 1
Reviewer 1 Report
An interesting work is conducted on the flexural creep response of hybrid GFRP-FRC sandwich panels. The authors have conducted a lot of experimental work and theoretical simulation analysis. More key results and findings are also reported in detail. However, the following comments should be considered to further improve the quality of the paper.
- The abstract part is well written with the clear logical thought. However, it is suggested that the authors add the research significance of flexural creep response, as well as the finding and results of creep properties of hybrid sandwich panels.
- Introduction:
(1) Line 31-33: “Nevertheless, the brittle failure, the high deformability, and……”, brittle fracture and high deformation capacity are contradictory. As known, the deformation capacity of fiber reinforced polymer composites is lower than that of steel. Therefore, the brittleness and low deformation capacity are the major disadvantages.
(2) The creep response of hybrid GFRP–concrete structures mainly depend on the creep responses of GFRP, polyurethane and concrete structures. As mentioned by the author, the research on the creep response of hybrid GFRP–concrete structures are very limited. Therefore, it is necessary to summarize the creep performances of GFRP, polyurethane and concrete structures, which should be related to the creep performance of hybrid structures. Please see some recent research work on the flexural creep response of GFRP and polyurethane-based composite. Composite Structures, 2022. 281: 115060. Polymers 2017, 9, 603. Construction and Building Materials, 2022, 314: 125587.
(3) The hybrid GFRP–concrete structure includes multiple interfaces among FRC, GFRP and PUR. Therefore, during the creep, will the above interfaces debond or peel off? It is suggested that the authors consider adding some research work on the effects of creep loading on the above interface damage.
(4) The summary of others’ research work should be simplified. For example, the summary of the research work [20-21] should be more concise, and the main finding, results, conclusions and solved/unsolved problems of the research instead of some details about the experimental should be analyzed and summarized. In addition, after summarizing the research work of others, the authors did not propose the importance and innovation of present research work. Please make necessary explanations and supplements.
(5) Why do you use GFRP as the bottom face sheet and lateral webs? Compared with GFRP, CFRP has better mechanical properties, creep and fatigue resistances and durability. Therefore, when CFRP is used in the above structures, the creep problem may be well solved.
- Part 2.1, the properties of FRC, GFRP, polyurethane and epoxy resin have been given in detail. Do the authors consider the interface performances between the above components? This may be critical to determine the damage and load bearing behavior of hybrid structure.
- For the flexural test with variable span (Figure 9), the samples go through several loading and unloading cycles. Why is the maximum load 15 kN? In addition, there seems to be a turning point for the load-displacement curve at the beginning of loading. What causes this phenomenon?
- All figures should remove the redundant scale on the right and top.
- In line 362, this title of part 3.2 is repeated with the title of part 3.1. Please check it.
- In part 3.2 (Table 8), the failure mode is generally that the concrete is crushed. What is the stress level of GFRP materials at this time? In addition, did the authors monitor the strain of the specimen along the different sections? The results of strain distribution may be more meaningful to analyze the failure mode.
- In figure 14, Why does the strain-time curve of GFRP present a perfect straight line? This may be inconsistent with the actual experimental test data. Why not show the experiment test data?
- The conclusion should be further simplified, including several key findings.
Author Response
Dear Reviewer, please find in the attached file, the replies and actions that were taken point-by-point to your queries.

Reviewer 2 Report
The paper is well written and represents a valuable contribution to the literature.
I recommend accepting the article in the present form.
Author Response

(The authors gave the same response as above.)

Reviewer 3 Report
The subject paper is interesting and its purpose complies with the journal’s aim and scope.
It presents creep tests, verified by analytical modelling of of hybrid GFRP-FRC sandwich panels. The results presented are original, interesting and advance current knowledge.
Lastly, in terms of language the manuscript is well written and requires only some polishing.
In greater detail, the following should be improved:
Introduction:
Line 45: in the not “the in”
Line 101 – 102: please rephrase
Materials …… and test methods:
Please provide main dimensions including thickness of each layer.
Add this in figure 1 as well
FRC: please give some background with respect to how this formulation was selected.
GFRP: please provide more details: eg how many stacks? Only later you show figure 3. Maybe add a reference to figure 3 and provide more details.
Lines 196 – 211: Please provide all necessary details of the parameters used in you GA model to ensure reproducibility of your results.
Line 350: Please correct: “On” average
Equations 5 and 6: please explain all parts of the equations
Author Response

(The authors gave the same response as above.)

Round 2
Reviewer 1 Report
The conclusion must be rewritten. The current writing is not the conclusion, but some experimental results. For the present research, at most 3-4 points are enough.
Author Response
The authors’ item-by-item replies to reviewers’ comments and actions taken are detailed in the attached file.

Reviewer 3 Report
The authors have effectively addressed all comments and the manuscript is suitable for publication.
Author Response

(The authors gave the same response as above.)
